# OverFill: Two-Stage Models for Efficient Language Model Decoding

**Woojeong Kim, Junxiong Wang, Jing Nathan Yan, Mohamed Abdelfattah,
Alexander M. Rush**

Cornell University
{wk247, jw2544, jy858, Mohamed, arush}@cornell.edu

## Abstract

Large language models (LLMs) excel across diverse tasks but face significant deployment challenges due to high inference costs. LLM inference comprises *prefill* (compute-bound) and *decode* (memory-bound) stages, with decode dominating latency particularly for long sequences. Current decoder-only models handle both stages uniformly, despite their distinct computational profiles. We propose OverFill, which decouples these stages to optimize accuracy-efficiency tradeoffs. OverFill begins with a full model for prefill, processing system and user inputs in parallel. It then switches to a dense pruned model, while generating tokens sequentially. Leveraging more compute during prefill, OverFill improves generation quality with minimal latency overhead. Our 3B-to-1B OverFill configuration outperforms 1B pruned models by 83.2%, while the 8B-to-3B configuration improves over 3B pruned models by 79.2% on average across standard benchmarks. OverFill matches the performance of same-sized models trained from scratch, while using significantly less training data. Our code is available at https://github.com/friendshipkim/overfill.

## 1 Introduction

Large language models (LLMs) have achieved remarkable success on a broad spectrum of tasks, from question answering to code generation. Yet, their massive parameter counts pose significant challenges for practical deployment, with *inference* emerging as a chief bottleneck. In modern LLMs, inference typically comprises two stages: *prefill* and *decode*. The prefill stage, where all input tokens are processed in parallel to build a Key-Value (KV) cache, is usually *compute-bound*: the performance is primarily limited by the utilization of computational units. The subsequent decode stage is *memory-bound*, where it generates each output token autoregressively. The main bottleneck here is repeatedly loading the model's large feed-forward (FFN) layers into memory.

However, current LLM architectures do not exploit the distinct computational characteristics of these two stages. Our motivation stems from the distinct computational profiles of the prefill and decode stages, which have led to a growing trend of disaggregating them. The first line of work (Zhong et al., 2024; Patel et al., 2024) is system-oriented, focusing on stage-specific resource allocation and parallelism while still using a single model across both stages. The second line of work (algorithmic-oriented) (Nair et al., 2024; Bergner et al., 2024) explores using models of different sizes for each stage, but often requires complex frameworks or delivers only small accuracy improvements. The question of interest is how to *decouple* the prefill and decode stages to achieve a stronger balance of accuracy and efficiency.

We propose **OverFill**, a two-stage system that dedicates maximal capacity to the prefill stage, while pruning the costly decode stage to reduce parameter loading. Specifically, the larger model is used only once to process the user input into a vector representation, and a smaller, pruned model subsequently handles token-by-token generation. As a result, OverFill drastically cuts the memory footprint and latency of decoding, especially for longer

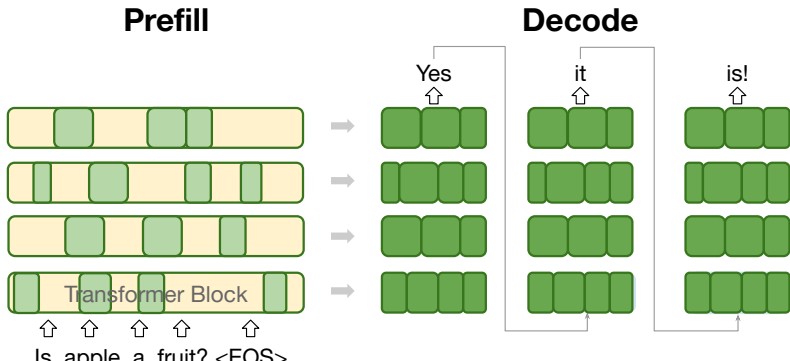

Figure 1: Overview of OverFill. OverFill uses the full model for prefill and a pruned model for sequential decoding. The yellow blocks represent the full model used for prefill. The decoder is initialized by selecting important channels (green blocks) from the full model. The blocks on the decoder side are in darker shades because they are updated during OverFill training. However, the full prefill model is kept frozen.

sequences. While it incurs a small prefill overhead compared to the standalone pruned model, this difference in cost is negligible compared to the dominant decoding latency. Moreover, OverFill starts with a single model and prunes it to build a smaller decoder, eliminating the extensive process of aligning two distinct models. Since only the small decoder is updated during training, OverFill is compatible with the original full model during serving. Importantly, OverFill is end-to-end trainable and does not require additional modules, allowing it to be optimized like any standard transformer architecture. Figure 1 illustrates the overview of our framework.

We validate OverFill in diverse decoder scales and compare OverFill accuracy on tasks such as question answering, math, and chain-of-thought reasoning. In a 3B-to-1B configuration, OverFill outperforms standalone pruned models by 83.2% and instruction-tuned model by 52.6% in average. OverFill matches or even outperforms similarly sized models trained from scratch, while using significantly fewer training tokens. We also demonstrate OverFill is pareto-optimal in both scales. OverFill achieves significant accuracy gain over standalone small models while posing minimal latency overhead. These efficiency gains become even more pronounced when long outputs or multiple candidates are generated, as the pruned model remains active throughout the autoregressive process.

## 2 Related work

**Model compression.** Pruning enhances the efficiency of LLMs by removing model components that contribute the least to the output. Structured pruning eliminates entire groups of parameters, such as channels, attention heads (Dery et al., 2024; Ma et al., 2023; Xia et al., 2023b; Ashkboos et al., 2024) or layers (Men et al., 2024; Yang et al., 2024; Kim et al., 2024). Structured pruning results in a more compact model while preserving the underlying architecture, keeping it hardware-friendly. In this work, we adopt width pruning (Dery et al., 2024; Ma et al., 2023; Xia et al., 2023b; Ashkboos et al., 2024), which preserves accuracy better than depth pruning. We specifically use the approach from Sreenivas et al. (2024) but without the additional KL loss term, focusing on optimizing pruning to balance performance and computational efficiency.

Quantization (Frantar et al., 2022; Lin et al., 2024; Xiao et al., 2023a) is another effective model compression method. Standard quantization methods accelerate both prefill and decode. In our two-stage decoding process, we can also consider high-precision prefill and low-precision decode, which we leave for future work.

**Decoding targeted speedups.** Various methods tackle the serial bottleneck in LLM decoding. Speculative decoding (Leviathan et al., 2023) leverages available compute to propose tokens in parallel using a small draft model. Researchers have explored specialized draft models (Sun et al., 2021; Xia et al., 2023a) and subnetworks of the target model (Schuster et al., 2022; Elhoushi et al., 2024; Zhang et al., 2023a; Liu et al., 2024a; Ankner et al., 2024). Among these, Du et al. (2024); Li et al. (2024) are particularly relevant to our work as they reuse target model representations to enhance drafting. Unlike speculative decoding, our approach eliminates rollbacks and calls the large model only once during prefill, avoiding parallel execution with the small model during decoding. This significantly reduces memory usage. We provide a theoretical analysis in the Appendix.

**KV cache compression.** Many studies have explored KV cache compression to address memory bottlenecks with heavy batching and long contexts, using methods like token eviction (Xiao et al., 2023b; Zhang et al., 2023b; Adnan et al., 2024), quantization (Sheng et al., 2023; Liu et al., 2024b), and prompt compression (Pan et al., 2024; Wingate et al., 2022). Our approach targets scenarios where loading weights is the primary memory bottleneck. While KV cache can dominate memory usage in certain scenarios, this typically occurs only at very long sequence lengths when using smaller models. For instance, in a 7B parameter model with batch size 4, model weights is the primary bottleneck up to 5K tokens (Adnan et al., 2024). Several works target the opposite challenge as ours: reducing prefill costs for very long contexts, where prefill becomes costly. These methods include architectural modifications (Sun et al., 2024), chunking (Zeng et al., 2024), token dropping (Fu et al., 2024), and prompt packing (Zhao et al., 2024). Notably, such KV cache compression and prefill acceleration approaches can be applied on top of our method for further optimization.

## 3 Method

We are interested in the setting of continual training of LLMs targeting instruction-tuning. In this setting we assume we have a large number of supervised examples of the form $(\mathbf{x}, \mathbf{y})$ where $\mathbf{x} = (x_1, x_2, \ldots, x_M)$ and $\mathbf{y} = (y_1, y_2, \ldots, y_N)$ are sequences of tokens, assumed for simplicity to be of a fixed length. We are particularly focused on $N > M$ since the model may use methods like chain-of-thought to answer problems.

Formally LLMs model the probability of a sequence $\mathbf{y}$ in a conditional autoregressive manner: $P(\mathbf{y} \mid \mathbf{x}) = \prod_{t=1}^{N} P(y_t \mid y_{<t}, \mathbf{x}; \theta)$, where $x_{<t}$ denotes all tokens preceding $x_t$, where $\theta$ is the model. The core probability of $P(y_t \mid y_{<t}, \mathbf{x}; \theta)$ is defined as a function of the Transformers cached hidden state,

$$P(y_t \mid y_{<t}, \mathbf{x}; \theta) \propto f(\text{Cache}([y_{<t}, \mathbf{x}]))$$

Where Cache is defined recurrently for any sequence of tokens $\mathbf{a}, \mathbf{b}$ as,

$$\text{Cache}([\mathbf{a}, \mathbf{b}]) = \text{Transformer}(\mathbf{a}, \text{Cache}(\mathbf{b}); \theta).$$

Due to this cache structure, when sampling from a language model, the computation happens in two-stages, prefill and decode. During prefill, the primary work is computing,

$$\mathbf{h}_{\text{pre}} \leftarrow \text{Transformer}(\mathbf{x}, \varnothing; \theta),$$

which can be done in parallel for all $\mathbf{x} = (x_1, x_2, \ldots, x_M)$. This stage is generally compute bound, since $\theta$ can be loaded once and compute is parallelized across $M$.

During decode, we autoregressively sample each $y_t$ and recurrently update,

$$\mathbf{h}_{\text{dec}_t} \leftarrow \text{Transformer}(y_t, \text{Cache}(\mathbf{h}); \theta).$$

This stage has a serial dependency in that it requires previously generated tokens in order to update the cache. As such, it is memory-bottlenecked in terms of speed as it needs to reload in $\theta$ at each step and cannot fully use available parallel compute. Our primary goal will be to speed this stage up in practice by reducing the effective size of the $\theta$ during the decode stage.

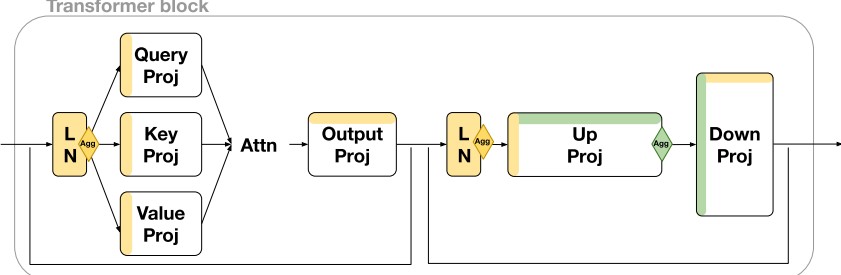

Figure 2: Width pruning strategy: Diamonds indicate activation aggregation points for measuring channel importance. Yellow represents the hidden dimension, while green denotes the intermediate dimension.

Finally, to train instruction models, we simply maximize the likelihood of each $(\mathbf{x}, \mathbf{y})$ instance. During this stage, we do not maximize the probability of the conditioning term $\mathbf{x}$ but only of the generated $\mathbf{y}$.

### 3.1 OverFill

We propose a simple approach of using the model's capacity asymmetrically during the prefill and decode stage. The approach uses a better model at prefill time to better utilize (overfill) the same cache of a smaller model. In our method, the same LLM parameters are used in two configurations:

- **Full Parameters** ($\theta$) for processing prefill.
- **Pruned Parameters** ($\theta' \subset \theta$) a subset of the full parameters for processing decode.

Our contribution modifies both training and inference procedures so that the tokens $\mathbf{x}$ use the full network and the tokens $\mathbf{y}$ use a pruned sub-network. Formally, we define a single set of model parameters but allow two "modes" (full vs. pruned), or more directly

$$\mathbf{h}_{\text{pre}} \leftarrow \text{Transformer}(\mathbf{x}, \varnothing; \theta),$$

$$\mathbf{h}_{\text{dec}_t} \leftarrow \text{Transformer}(y_t, \text{Cache}(\mathbf{h}); \theta').$$

Upon deciding on the prune subset to use, we freeze the full model and train only the pruned model with standard teacher forcing on the output tokens $\mathbf{y}$. This approach ensures that the full model retains its well-initialized weights while accelerating training. Also, it allows the pruned decoder to be seamlessly integrated into existing models.

### 3.2 Compatible pruning

In order for OverFill to be a compatible method for LLM generation, it requires two aspects: (1) The pruned model must be significantly smaller than the original $|\theta'| \ll |\theta|$ and (2) they must have a compatible cache representation $\mathbf{h}$. In Transformers, the KV Cache representation $\mathbf{h}$ is $\mathbb{R}^{2 \times L \times D}$ per sequence length, where $D$ is the embedding dimension of the Transformer keys and values, and $L$ is the number of layers. While there are many different pruning methodologies that fit criteria (1), e.g. depth pruning and unstructured pruning, we target methods that can maintain criteria (2). Specifically we utilize a targeted form of width pruning, that avoid changing the cache dimension.

Our approach is based on the static channel pruning strategy from Sreenivas et al. (2024). Figure 2 gives a schematic overview of the pruning template. Following the standard notation for transformers for the attention projections and FFNs we define our pruned parameters $\theta'$ as,

$$\mathbf{W}^q, \mathbf{W}^k, \mathbf{W}^v \in \mathbb{R}^{D' \times D}, \quad \mathbf{W}^o \in \mathbb{R}^{D \times D'}, \quad \mathbf{W}^\uparrow \in \mathbb{R}^{D' \times 4D'}, \quad \mathbf{W}^\downarrow \in \mathbb{R}^{4D' \times D'}$$

| Full ($|\theta|$) | Pruned ($|\theta'|$) | Pruning ratio ($P$) | Hidden dim. ($D$) | Layers ($L$) |
|---|---|---|---|---|
| 3.21B | 0.52B | 0.7 | 921 | 28 |
| 3.21B | 1.24B | 0.45 | 1689 | 28 |
| 3.21B | 2.01B | 0.25 | 2304 | 28 |
| 8.03B | 3.19B | 0.43 | 2334 | 32 |
| 14.76B | 7.62B | 0.43* | 2944 | 48 |

Table 1: Model sizes and width pruning configurations. *For the 14B model, we use a hardware-friendly configuration by default, where different pruning ratios are applied to the hidden dimension and the intermediate dimension. See Table 8.

This new network keeps a subset $D' < D$ for each of the weight matrices in the network. Layer norm and embedding parameters are defined similarly.

To obtain the best starting compact sub-network, we first pick the pruning ratio $P = 1 - D'/D$ and then select the best rows and columns. To decide on what to prune, we pass a small calibration set through the model and aggregate activations as in Figure 2 at three points per layer: 1) before the attention projection, 2) before the FFN, and 3) within the FFN. After aggregation we end up with a tensor of shape batch by sequence length by dimension. We reduce this tensor using the L2 norm across the batch, and the mean across the sequences to derive an importance scores for each dimension. Finally, we retain the top $(1 - P)\%$ of channels to calculate $\theta'$.

As mentioned above, this pruning acts as a starting point for determining the shape of $\theta'$. Once determining this shape, additional finetuning is run on set of instruction examples to adjust the weights of the parameters to this new setting.

# 4 Experimental setup

## 4.1 Data

We use two instruction-tuning datasets for training: OpenHermes-2.5 and Infinity-Instruct. For main experiments, we adopt Infinity-Instruct (BAAI, 2024) with 7M instances. We filter out non-English data using the provided language tags. For pruning ratio sweeps, we use OpenHermes-2.5 (Teknium, 2023) which has 1M instances, of which 997k are used for training and 3k for validation. The total training tokens amount is 38M for OpenHermes-2.5 and 212M for Infinity-Instruct.

These datasets are well-suited for our task due to their natural separation between context and input. Both datasets are formatted with distinct tags: System, User, and Assistant. The System and User parts are concatenated to the context, which is processed by the full model. The Assistant part serves as the target output to be predicted by the pruned model. To construct the model inputs, we inherit each dataset's original chat template.

## 4.2 Model

We evaluate our method on three base models in two model families: Llama 3.2-3B-Instruct, Llama 3.1-8B-Instruct (Dubey et al., 2024), and Qwen 2.5-14B-Instruct (Team, 2024). For pruning, we apply the strategy outlined in Section 3.2. We do not prune attention heads or layers to preserve the dimensionality of the KV cache and retain as much information as possible passed to the pruned decoder. Table 1 presents the full and pruned model sizes along with their pruning configurations. We adopt the training hyperparameters from Tunstall et al., as presented in Table A.1

| Model | Pruned* | OverFill * | 1B-Tuned* | 1B-Inst | 3B-Inst |
|---|---|---|---|---|---|
| Decoder Size | 1.2B | 1.2B | 1.2B | 1.2B | 3.2B |
| GSM8K-CoT | 45.4 (±1.4) | **59.2 (±1.4)** | 47.6 (±1.4) | 45.7 (±1.4) | 78.4 (±1.1) |
| ARC | 36.4 (±1.4) | **77.8 (±1.2)** | 42.5 (±1.4) | 56.5 (±1.5) | 78.2 (±1.2) |
| MMLU | 33.7 (±0.4) | **63.7 (±0.4)** | 38.7 (±0.4) | 47.9 (±0.4) | 63.3 (±0.4) |
| MATH | 6.1 (±0.3) | **8.3 (±0.4)** | 5.2 (±0.3) | 16.7 (±0.5) | 35.0 (±0.6) |
| WMT16-DE-EN | 14.7 (±0.3) | **31.4 (±0.5)** | 28.0 (±0.4) | 29.7 (±0.4) | 36.9 (±0.4) |
| IFEval | 26.4 (±1.9) | **44.2 (±2.1)** | 26.1 (±1.9) | 48.1 (±2.2) | 69.5 (±2.0) |
| NQ | 5.2 (±0.4) | **12.1 (±0.5)** | 7.8 (±0.5) | 10.8 (±0.5) | 19.7 (±0.7) |
| MMLU-Redux | 26.06 | **40.93** | 27.29 | 18.21 | 56.95 |
| CRUX | 8.62 | **8.75** | 4.88 | 9.00 | 25.71 |

Table 2: Results in 1B scale. Bold indicates the best models under the same training data regime. * means identically trained with the same data (less data compared to the Instruct models).

| Model | Pruned* | OverFill * | 3B-Tuned* | 3B-Inst | 8B-Inst |
|---|---|---|---|---|---|
| Decoder Size | 3.2B | 3.2B | 3.2B | 3.2B | 8.0B |
| GSM8K-CoT | 55.4 (±1.4) | **69.8 (±1.3)** | 61.8 (±1.3) | 78.4 (±1.1) | 84.6 (±1.0) |
| ARC | 42.5 (±1.4) | **83.4 (±1.2)** | 61.7 (±1.4) | 78.2 (±1.2) | 83.4 (±1.1) |
| MMLU | 35.8 (±0.4) | **69.4 (±0.4)** | 48.3 (±0.4) | 63.3 (±0.4) | 69.4 (±0.4) |
| MATH | 7.2 (±0.4) | 15.1 (±0.5) | **17.3 (±0.5)** | 35.0 (±0.6) | 36.2 (±0.7) |
| WMT16-DE-EN | 18.5 (±0.3) | **35.6 (±0.5)** | 25.8 (±0.5) | 36.9 (±0.4) | 41.0 (±0.4) |
| IFEval | 28.4 (±1.9) | **44.0 (±2.1)** | 32.3 (±2.0) | 69.5 (±2.0) | 73.9 (±1.9) |
| NQ | 8.0 (±0.5) | **14.5 (±0.6)** | 5.4 (±0.4) | 19.7 (±0.5) | 19.1 (±0.7) |
| MMLU-Redux | 28.80 | 43.12 | **43.95** | 56.95 | 61.66 |
| CRUX | 6.12 | **27.00** | 24.88 | 25.71 | 39.38 |

Table 3: Results in 3B scale. Bold indicates the best models under the same training data regime. * means identically trained with the same data (less data compared to the Instruct models).

## 4.3 Downstream evaluation

We evaluate OverFill on downstream generation tasks, including math, code, question answering, and machine translation, using the LM Eval Harness (Gao et al., 2024). Details on evaluation metrics and the number of few-shot examples are provided in Table 6. We use generation-based evaluation for all tasks, including multiple-choice question answering, whereas an alternative approach is to compare the probability of answer choices. To assess longer-form generation, we use MMLU-Redux and CRUXEval from the ZeroEval benchmark (Lin, 2024). ZeroEval is designed for evaluating instruction-tuned models, with MMLU-Redux focusing on general knowledge reasoning and CRUXEval assessing code reasoning, understanding, and execution capabilities. Models are prompted to provide both detailed reasoning steps and final answers in a JSON-formatted output. We use greedy decoding for all generations.

# 5 Results

## 5.1 Accuracy

Table 2 presents the downstream evaluation results at the 1B scale. OverFill begins with a 3B model and then pruned to a 1B-scale model for decoding. Our roofline model, Llama-3B-Instruct, handles both prefill and decoding.

| Model | Pruned* | OverFill * | 7B-Tuned* | 7B-Inst | 14B-Inst |
|---|---|---|---|---|---|
| Decoder Size | 7.6B | 7.6B | 7.6B | 7.6B | 14.8B |
| GSM8K-CoT | 75.3 (±1.2) | 78.1 (±1.1) | **81.0 (±1.1)** | 81.7 (±1.1) | 75.9 (±1.1) |
| ARC | 70.8 (±1.3) | **90.6 (±0.9)** | 83.0 (±1.1) | 89.5 (±0.9) | 90.6 (±0.9) |
| MMLU | 51.2 (±0.4) | **77.9 (±0.3)** | 65.3 (±0.8) | 72.2 (±0.4) | 77.9 (±0.3) |
| WMT16-DE-EN | 32.0 (±0.3) | **38.2 (±0.4)** | 37.6 (±0.4) | 37.5 (±0.4) | 38.6 (±0.4) |
| IFEval | 40.5 (±2.1) | **51.0 (±2.1)** | 32.2 (±2.0) | 71.5 (±2.0) | 78.3 (±1.7) |

Table 4: Results in 7B scale. Bold indicates the best models under the same training data regime. * means identically trained with the same data (less data compared to the Instruct models).

We compare accuracy across models trained under the same 1B decoder size and data regime. One baseline is the standalone pruned model, derived from the same original model, where a single model performs both prefill and decoding. Additional baselines include Llama 1B variants: (1) Llama 1B-base model finetuned on the same data as our pruned models and (2) Llama 1B-Instruct, a highly optimized model with more extensive instruction tuning. We observe that further tuning of Llama-Instruct consistently hurts performance across all tasks, as shown in Table 7. Therefore, we focus on comparisons with untuned Instruct models.

Our results highlight the following: OverFill consistently outperforms the standalone pruned model and finetuned 1B-Base model across all tasks by a significant margin, demonstrating the effectiveness of the two-stage approach. OverFill shows accuracy improvements on both multiple-choice tasks, such as ARC-Challenge (Clark et al., 2018) and MMLU (Hendrycks et al., 2020), which typically involve short generations, and tasks requiring longer responses, such as GSM8K (Cobbe et al., 2021), MMLU-Redux (Gema et al., 2024), and CRUXEval (Gu et al., 2024). This demonstrates that smart prefill benefits not only tokens close to it but also those generated later. When generation is short, OverFill can even match the full performance of the roofline model while maintaining a lower decoding cost. OverFill also outperforms Llama 1B-Instruct on 7 out of 9 tasks. This is particularly notable given that Llama 1B-Instruct is a highly optimized model at this scale, likely benefiting from more extensive data than our approach.

A similar trend is observed for 3B-scale models (Table 3). Here, OverFill with an 8B prefill and 3B decode outperforms the pruned model on all tasks and the trained 3B-Base model on 7 out of 9 tasks. The 3B-Instruct model is a strong compact baseline, with performance close to that of 8B models, yet OverFill matches it on several tasks. This pattern also extends to the Qwen2.5 family and a larger scale (14B to 7B). OverFill with a 7B decoder outperforms both the pruned model and the trained 7B-Base model on most tasks.

## 5.2 Efficiency

We show OverFill poses minimal overhead to the small standalone model by presenting end-to-end latency in 1B and 3B scales. We use vLLM (Sreenivas et al., 2024) v0.8.5 for benchmarking and separately measure prefill and decode latency. All experiments are conducted on a single NVIDIA A100 GPU. We slightly modify OverFill configuration to better leverage NVIDIA Tensor Cores, which is optimized for matrix multiplications in tiles (typically 16x8 or 16x16). For a fair comparison, we maintain a larger parameter size than the one used for accuracy evaluation. The exact configurations are provided in Table 8. We report the mean and standard deviation of 10 runs with 2 warm-ups.

Figure 3 presents latency across varying generation lengths, with fixed prompt length and batch size. The results show that OverFill asymptotically reaches the runtime of a small model and this trend becomes more pronounced in longer generations, where decoding cost dominates over prefill cost. At the 1B scale, both Pruned-1B and OverFill-1B exhibit higher latency compared to Llama-1B, as they have more transformer blocks, and transformers are more efficiently parallelized along width rather than depth. This suggests a limitation of

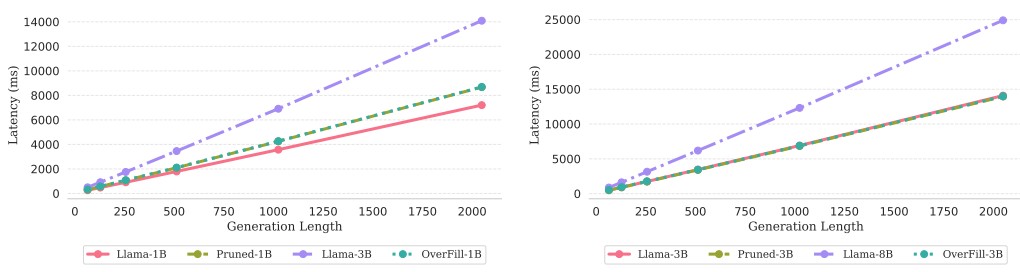

Figure 3: Latency across different generation lengths. Prompt length is fixed to 128 with batch size 4.

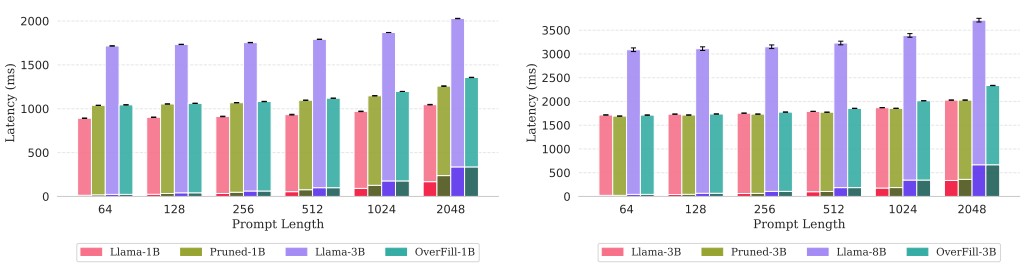

Figure 4: Latency under different prompt lengths, with generation length fixed at 128 and batch size set to 4. Prefill and decode latencies are shown as stacked bars, with darker shades representing prefill and lighter shades representing decode.

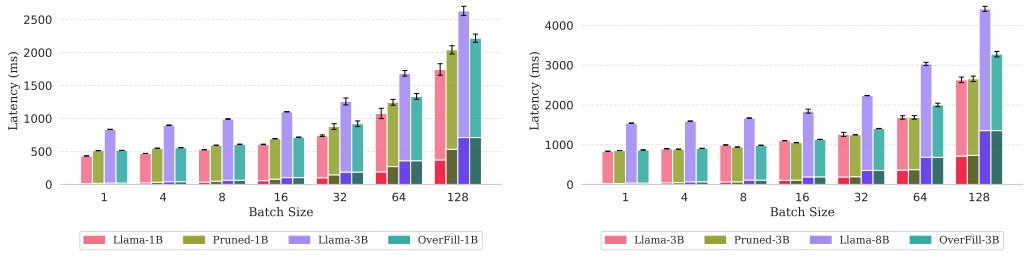

Figure 5: Latency under different batch sizes. Prompt length and generation length are fixed to 128.

width pruning, highlighting the need for a more balanced decoder architecture to improve both efficiency and accuracy. Still, both remain significantly faster than the full 3B model. At the 3B scale, where models have similar depth, Llama-3B, Pruned-3B, and OverFill-3B show nearly identical latency. Overall, OverFill achieves higher accuracy while introducing minimal latency overhead.

We measure latency while varying prompt lengths with a fixed generation length, as shown in Figure 4. Decoding latency remains the dominant factor over prefill latency in all cases, even when the prompt length is 16 times the generation length. Across all prompt lengths and model scales, OverFill consistently achieves speedups compared to the full model.

We also sweep batch size to account for diverse serving environments. The results are shown in Figure 5, where prompt and generation lengths are the same. In small batch scenarios, OverFill introduces minimal latency overhead to the pruned models. However, as batch size increases, prefill becomes relatively more expensive as decoding shifts from being memory-bound to compute-bound.

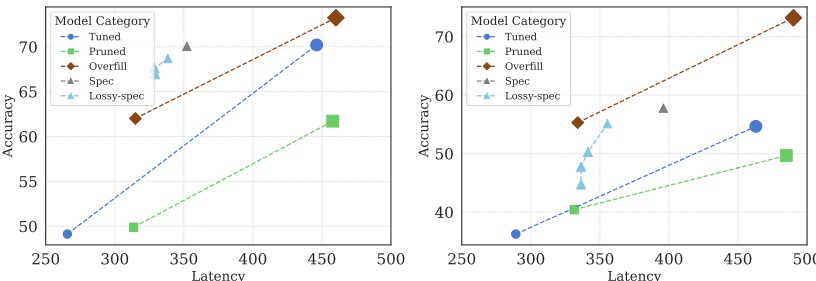

Figure 6: Latency–accuracy tradeoff on GSM8K with temperature 0.01 (left) and temperature 1 (right). Models trained with the same approach are connected by dashed lines, and dot size indicates the corresponding decoder size.

# 6 Analysis

## 6.1 Pareto optimality

We demonstrate that OverFill is Pareto optimal compared to models under the same training regime. Figure 6 plots end-to-end latency versus accuracy on GSM8K with Chain-of-thought (CoT). CoT is a standard practice that prompts models to reason step-by-step before producing a final answer, thereby boosting accuracy but resulting in longer generations. In our experiments, the average prompt length is 613 tokens with 4 demonstrations, and the average generation length is 120 tokens. Latency is measured using vLLM on a single NVIDIA H100 GPU. We fixed the batch size to 1 across all experiments. OverFill achieves Pareto-optimality compared to finetuned and pruned models. Interestingly, Pruned models underperform compared to finetuned-Llama counterparts, likely because pruned architectures require extensive adaptation (Sreenivas et al., 2024) to recover their accuracy.

## 6.2 Comparison to speculative decoding

Speculative decoding and OverFill are similar in that both decode with the assistance of a smaller model. We present speculative decoding latency results in Figure 6. The original speculative decoding method (Xia et al., 2023a) is theoretically lossless, meaning it exactly matches the target model's output. We also evaluate a lossy variant with lenient rejection sampling (Zhou et al., 2023b), in which more tokens proposed by the draft model are accepted to increase speedup. In our setup, the target model is a finetuned 3B model and the draft model is a finetuned 1B model using the same data for both. We measure latency with a single batch size, which is the most common setting for speculative decoding benchmarks.

Under temperature-1 decoding, OverFill achieves a better tradeoff. OverFill 3B–1B attains a 1.06× speedup over lossy speculative decoding at the same accuracy level. However, in near-greedy decoding (low temperature), speculative decoding performs particularly well because the outputs of the target and draft models are already very similar. As a result, lenient rejection sampling provides limited additional benefit in this regime, since the acceptance rate is naturally high.

In more realistic serving scenarios with multiple concurrent requests, OverFill shows greater potential. Using the same vLLM setup with a maximum batch size of 256, our pruned decoder achieves a throughput of 7,913 tokens/s, compared to 2,871 tokens/s for standard speculative decoding. This advantage comes from our approach accepting all tokens from the drafter without rejection sampling, allowing more tokens to be emitted per second. While recent speculative decoding variants have been proposed to improve throughput (Miao et al., 2024; Sadhukhan et al., 2024), we leave comparisons to these methods for future work.

## 6.3 Impact of pruning ratio

We vary the pruning ratio while keeping the full model (3B) fixed to examine how the capacity gap between the full and pruned models affects the benefits of OverFill. As shown

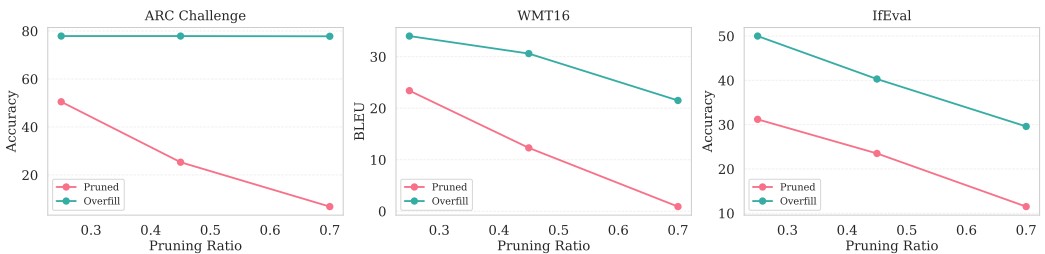

Figure 7: Performance comparison between Pruned and OverFill across three tasks varying pruning ratios.

in Fig. 7, OverFill consistently maintains significantly higher accuracy than the pruned model with the same decoder configuration. Interestingly, in ARC, where most information is processed during the prefill stage, a lightweight decoder with a pruning ratio of 0.7 shows no performance degradation. However, in translation and instruction-following tasks, both Pruned and OverFill experience performance drops with increased pruning. In some cases, OverFill degrades at a slower rate than Pruned or follows a similar trend as more channels are pruned.

### 6.4 Accuracy by generation length

We observe that the benefits of OverFill persist for long generations. Figure 8 shows the probabilities assigned to correct tokens across their absolute positions in the output. The results show that OverFill consistently predicts tokens more accurately than the pruned model, demonstrating that the advantage of smart prefill extends to long generations.

However, as the distance from the prefill grows, the gap gradually narrows, as the generation becomes more dependent on the smaller decoder. This suggests that while OverFill may not be optimal for extremely long generations, it remains highly effective for many practical use cases, such as bootstrapping multiple generations during testing. We plan to further explore whether periodically refreshing the prefill can help maintain its benefits uniformly throughout the entire sequence.

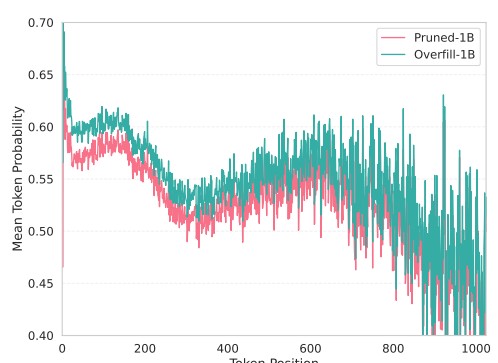

Figure 8: Probability assigned to the correct token up to the 1024-th position, averaged across the sampled validation set.

## 7 Conclusion & Future work

This work presents a method for improving LLM generation with minimal latency increase by using a compatibly pruned subset of parameters for memory-bound decoding while retaining the full model for compute-bound prefill. We show that this approach outperforms fine-tuning base models of the same size and a standalone pruned model, with only minimal latency slowdowns. Our method is one of many possible strategies for training compatible pruned decoders and we believe there is a large design space of other architectures. For instance, pruning attention could further optimize KV cache size and decoding efficiency. Scaling this approach with larger training could also extend its benefits to even more memory-constrained models.

## Acknowledgment

AMR was supported by NSF CAREER 2037519. This research is supported in part by the Office of the Director of National Intelligence (ODNI), Intelligence Advanced Research Projects Activity (IARPA), via the HIATUS Program contract #4202696884. We thank Yueying Li (Cornell Tech), Celine Lee (Cornell Tech), Ben Athiwaratkun (Together AI), and Muru Zhang (Together AI) for helpful discussions and feedback.

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

# A Appendix

## A.1 Training hyperparmeters

| LR | LR scheduler | Warmup | Max seq length |
|---|---|---|---|
| 2e-05 | cosine | 0.01 | 2048 |

Table 5: Training hyperparameters

## A.2 Downstream evaluation details

| Task | Metric | Few-shot |
|---|---|---|
| GSM8K (Cobbe et al., 2021) | Accuracy | 4 |
| ARC-challenge (Clark et al., 2018) | Accuracy | 0 |
| MMLU (Hendrycks et al., 2020) | Accuracy | 4 |
| MATH (Hendrycks et al., 2021) | Accuracy | 4 |
| WMT16 (Bojar et al., 2016) | BLEU | 4 |
| IfEval (Prompt-level) (Zhou et al., 2023a) | Accuracy | 4 |
| Natural Questions (Kwiatkowski et al., 2019) | F1 | 4 |
| MMLU-Redux (Gema et al., 2024) | Accuracy | 0 |
| CRUXEval (Gu et al., 2024) | Accuracy | 0 |

Table 6: Evaluation details.

## A.3 Finetuning Instruct models

| Model | GSM8K | ARC | MMLU | MATH | WMT16 | IfEval | NQ |
|---|---|---|---|---|---|---|---|
| 1B-Instruct | 45.7 | 56.5 | 47.9 | 16.7 | 29.7 | 48.1 | 108 |
| 1B-Instruct-Tuned | 40.8 | 49.4 | 42.4 | 6.9 | 28.9 | 34.4 | 3.1 |
| 3B-Instruct | 78.4 | 78.2 | 63.3 | 36.9 | 78.5 | 69.5 | 19.7 |
| 3B-Instruct-Tuned | 64.4 | 70.6 | 55.7 | 12.8 | 34.2 | 53.2 | 8.9 |

Table 7: Finetuning Llama-Instruct models.

## A.4 Speed benchmark model configurations

| Model | Hidden dim. | Intermediate dim. | Layers | Params. |
|---|---|---|---|---|
| 1B-Pruned | 1689 | 4505 | 28 | 1.24B |
| 1B-Pruned-standard | 1792 | 4096 | 28 | 1.26B |
| 3B-Pruned | 2334 | 8171 | 32 | 3.19B |
| 3B-Pruned-standard | 2432 | 7680 | 32 | 3.21B |
| 7B-Pruned-standard | 2944 | 11776 | 48 | 7.62B |

Table 8: Speed benchmark model configurations.

