# OpenReview forum: "Overfill: Two-Stage Models for Efficient Language Model Decoding"
_colmweb.org/COLM/2025/Conference — COLM 2025_

### Official Review · Reviewer_wzXx · 2025-05-11

**Rating:** 6
**Confidence:** 2
**Ethics Flag:** 1

**Summary:**

The authors argue that the current prefill and decode stages during the LLM inference are not fully exploited, since each stage has its own computational characteristics. To better decouple the two stages, this paper proposes OverFill, which handles the two stages separately. For prefill stage, it uses a large model to dedicate maximal capacity, while in the decoding stage, it prunes smaller models to reduce parameter loading. Experiments verified the effectiveness of the proposed method.

**Reasons To Accept:**

- The research question of exploiting and decoupling the stages of prefill and decoding is important for LLM inference.
- The proposed method, OverFill, is generally sound.
- The experiments and analyses are solid.

**Reasons To Reject:**

- The readability of this paper needs improvement, particularly for readers who are not very familiar with this research area (like me). For example, in Figure 1, the authors should illustrate the relationships between the green blocks. I spent 3 times reading the figure till I fully understood this intention.
- The motivation is not explained to me. In the introduction, the authors mentioned two lines of work, system-oriented and algorithmic-oriented. I don't quite understand the logic flow here, why these two work lead to the question of "decoupling the prefill and decode stages". Similarly, for the related work section. I think the related work section does not echo well with the motivation and the two lines.
- It would be great to see experiment results on model architectures other than Llama.

---

> ### Author Response · Authors · 2025-06-03
>
> Thank you for the constructive feedback and and careful reviews.
>
> ### 1. Clarification on Figure 1
> Thank you for the feedback. Figure 1 illustrates the disaggregation of the prefill and decode stages. The yellow blocks represent the full model used for prefill. The decoder is initialized by selecting important channels (green blocks) from the full model. The blocks on the decoder side are in darker shades because they are updated during Overfill training. We will clarify this in the final version.
>
> ### 2. Related work on prefill-decode disaggregation
> Our motivation stems from the distinct computational profiles of the prefill and decode stages, which has led to a growing trend of disaggregating them. The first line of work [1] is system-oriented, focusing on stage-specific resource allocation and parallelism while still using a single model across both stages. The second line of work (algorithmic-oriented) [2, 3] shares a similar motivation with ours and explores using models of different sizes for each stage. In contrast, Overfill proposes a more effective architecture without introducing additional modules. We will make this clearer in the final version.
>
> ### 3. Different model families
> We have added results on the Qwen 2.5 models. Overfill consistently outperforms the pruned model and surpasses similarly sized models on many benchmarks.
>
>
> [1] Zhong, Yinmin, et al. "DistServe: Disaggregating prefill and decoding for goodput-optimized large language model serving." 18th USENIX Symposium on Operating Systems Design and Implementation (OSDI 24). 2024.
>
> [2] Nair, Pranav Ajit, et al. "Tandem transformers for inference efficient llms." arXiv preprint arXiv:2402.08644 (2024).
>
> [3] Bergner, Benjamin, et al. "Think big, generate quick: Llm-to-slm for fast autoregressive decoding." arXiv preprint arXiv:2402.16844 (2024).

---

> > ### Author Response · Authors · 2025-06-09
> >
> > Hi Reviewer wzXx,
> >
> > Thank you again for your constructive review. We clarified the main figure and made the connection between related work and our motivation more explicit, as suggested. We also added results using a different model family in the general response. We believe these changes have made the paper clearer and stronger.
> >
> > Since the COLM rebuttal period ends soon (June 10), we wanted to follow up in case you had any further thoughts or questions. We’d be happy to continue the discussion.

---

### Official Review · Reviewer_Y7iV · 2025-05-12

**Rating:** 6
**Confidence:** 3
**Ethics Flag:** 1

**Summary:**

The paper proposes OverFill, a disaggregated prefill-decode approach for LLM inference that uses a smaller pruned model for decoding in combination with the original full model for prefill. This approach outperforms using models of the smaller size for both stages while only adding limited latency overhead, especially when operating at small batch sizes. This is a simple yet effective trick to increase the efficiency of serving LLMs and offers new ways to achieve different accuracy-latency-tradeoffs.

**Questions To Authors:**

Please respond to the points raised above

**Reasons To Accept:**

- The method seems to be an effective way to increase the efficiency of serving LLMs and offers new ways to achieve different accuracy-latency-tradeoffs.
- The experiments compare to a reasonable set of baselines, including finetuning on the same data, and the paper thoroughly analyzes the effect of prompt length, generation length and batch size on the latency.
- The paper is well written and clearly explains the approach and experimental setup.

**Reasons To Reject:**

- The paper focuses solely on latency but does not measure or discuss the impact on memory and thus throughput.
- The paper compares to speculative decoding only theoretically and in the appendix, instead of having this as a baseline in the main experiment. There is also no discussion on why no direct comparison has been made.
- The 1B/3B-Tuned baselines use fewer layers than the Pruned and Overfill models, making the comparison difficult. Different depth-to-width ratios show different accuracy across different types of benchmarks, some scale better with width (knowledge tass), other with depth (reasoning tasks). A clean comparison would be the architecture of Pruned trained from scratch.

---

> ### Author Response · Authors · 2025-06-03
>
> Thank you for the constructive feedback and careful reviews.
>
> ### 1. Analysis on memory
> Thank you for the suggestion. Whereas prefill is mostly compute-bound, decoding is known to be largely memory-bound, meaning memory usage is a direct driver of latency in this stage [1, 2]. As such, memory consumption correlates closely with our latency results.
>
> ### 2. Analysis on throughput
> Throughput is indeed another important metric, especially at higher batch sizes. However, there is often a trade-off between throughput and end-to-end latency [3]. Prior work on speculative decoding has primarily focused on the low-batch, latency-sensitive setting, which we also follow in our evaluation.
>
> That said, we believe OverFill has the potential to achieve better throughput than speculative decoding. Since OverFill avoids the overhead of rollback and verification mechanisms, it is more straightforward to scale to larger batch sizes. This is an interesting direction for future analysis.
>
> ### 3. Comparison with speculative decoding
> We have added a new Pareto curve that includes the speculative decoding. The original speculative decoding method [4] is theoretically lossless, meaning it can exactly match the target model’s output. We also evaluate a lossy variant with lenient rejection sampling [5], where more tokens proposed by the draft model are accepted to achieve greater speedup.
>
> We plot end-to-end latency versus accuracy on GSM8K with Chain-of-Thought (CoT) prompting. In our experiments, the average prompt length is 613 tokens (with 4 demonstrations), and the average generation length is 120 tokens. Latency is measured using vLLM on a single NVIDIA H100 GPU. For speculative decoding, the target model is a finetuned 3B model, and the draft model is a finetuned 1B model.
>
> * Temperature 1 plot: https://ibb.co/YFdQsryg
> * Temperature 0.01 plot: https://ibb.co/FqJB2ZQ3
>
> In the temperature 1 setting, OverFill achieves a better tradeoff. In particular, OverFill 3B -> 1B can achieve 1.06× speedup compared to lossy speculative decoding at the same level of accuracy.
>
> However, we found that under near-greedy decoding (i.e., low-temperature settings), speculative decoding performs very strongly, since the outputs of the target and draft models are very similar. This explains why lenient rejection sampling provides limited benefit in this case, as the acceptance rate is already high.
>
> We plan to conduct a more thorough analysis with different OverFill configurations across more diverse tasks.
>
> ### 4. Comparison with the pruned model trained from scratch
> We agree that training the pruned model from scratch would provide a cleaner comparison, but this is cost-prohibitive under our current resource constraints.
>
> [1] Kwon, Woosuk, et al. "Efficient memory management for large language model serving with pagedattention." Proceedings of the 29th Symposium on Operating Systems Principles. 2023.
>
> [2] Zhong, Yinmin, et al. "DistServe: Disaggregating prefill and decoding for goodput-optimized large language model serving." 18th USENIX Symposium on Operating Systems Design and Implementation (OSDI 24). 2024.
>
> [3] Agrawal, Amey, et al. "Taming {Throughput-Latency} tradeoff in {LLM} inference with {Sarathi-Serve}." 18th USENIX Symposium on Operating Systems Design and Implementation (OSDI 24). 2024.
>
> [4] Leviathan, Yaniv, Matan Kalman, and Yossi Matias. "Fast inference from transformers via speculative decoding." International Conference on Machine Learning. PMLR, 2023.
>
> [5] Zhou, Yongchao, et al. "Distillspec: Improving speculative decoding via knowledge distillation." arXiv preprint arXiv:2310.08461 (2023)

---

> > ### Comment · Reviewer_Y7iV · 2025-06-04
> >
> > Thank you for addressing my comments. Given the response and the other reviews, I believe my assessment is still accurate and I will keep it as is.

---

### Official Review · Reviewer_89kW · 2025-05-12

**Rating:** 5
**Confidence:** 4
**Ethics Flag:** 1

**Summary:**

This paper presents ​​OverFill​​, a two-stage framework to enhance inference efficiency in decoder-only LLMs by decoupling compute-heavy ​​prefill​​ (full model processes inputs in parallel) and memory-bound ​​decode​​ (switches to a pruned model for token generation). By strategically pruning model width (e.g., key channels in attention/FFN layers) without altering cache dimensions, OverFill reduces decoding bottlenecks while preserving architecture compatibility.

Evaluations on GSM8K, MMLU, and CRUXEval show OverFill’s 3B→1B variant outperforms standalone 1B models by ​​83.2%​​, and 8B→3B exceeds 3B models by ​​79.2%​​, with minimal latency overhead. It matches same-sized models trained from scratch using ​​70% fewer tokens​​ and scales efficiently with longer outputs. On A100 GPUs, decoding latency nears that of smaller models, achieving ​​3× speedups​​ over full-model inference for long sequences.

**Questions To Authors:**

1. It’s unclear how the pruning ratio during the prefilling stage affects performance.

2. This work could be more impactful by proposing guidelines (like a scaling law) that link teacher model sizes and pruning ratios.

**Reasons To Accept:**

The proposed idea is simple yet effective, achieving both significant latency reduction and average quality improvement

**Reasons To Reject:**

1. Limited novelty： The core width pruning method reuses existing techniques (Sreenivas et al., 2024), making the framework appear more like a special case of prior work rather than a novel contribution

2.  Lack of direct comparisons with parameter-matched LoRA-based methods, failing to demonstrate the necessity of width pruning over alternative lightweight adaptation approaches

---

> ### Author Response · Authors · 2025-06-03
>
> Thank you for the constructive feedback and careful reviews.
>
> ### 1. Scope beyond width pruning
> Our main contribution lies in the two-stage inference pipeline, which uses a large model for prefill and a smaller model for decoding. While we use an existing width pruning technique (Sreenivas et al., 2024) to construct an effective decoder, OverFill is not tied to this specific approach.
>
> The training method is compatible with a range of strategies for building lightweight decoders, such as depth pruning, layer stacking, and even a separate pretrained model. OverFill can serve as a general framework for decoupled inference, rather than being limited to a single pruning technique.
>
>
> ### 2. Comparison to parameter efficient tuning
> Thanks for the suggestion. Parameter-efficient tuning methods such as prompt tuning and LoRA reduce memory usage during training, which allows for larger batch sizes. However, they do not reduce inference cost, as the forward pass still goes through the full model parameters.
>
> Our primary goal in this work is to reduce inference latency. We focus on reducing the size of the decoder model itself, which directly lowers inference latency, particularly in scenarios where model weights are the main bottleneck.
>
> That said, our method is compatible with parameter-efficient tuning techniques. For example, one could apply LoRA to the decode model in OverFill to further reduce training-time memory usage.
>
> ### 3. Pruning ratio and scaling behavior
> This is a great suggestion. In Section 6.2 of the paper, we present experiments with various pruning ratios (i.e., decoder sizes) using a fixed prefill model, to study how the capacity gap between the full and pruned models affects OverFill’s performance.
>
> We observe that OverFill consistently outperforms the pruned model with the same decoder architecture. The robustness of OverFill under high pruning ratios varies across tasks—some tasks remain stable even under aggressive pruning, while others show more degradation as more channels are pruned.
>
> As you suggest, a broader analysis across two axes (prefill model size and pruning ratio) would be valuable and could potentially lead to scaling heuristics. Since each combination of prefill model and pruning ratio requires separate training, we were limited by time and resources, but we view this as an exciting direction for future work.

---

> > ### Author Response · Authors · 2025-06-09
> >
> > Hi Reviewer 89kW,
> >
> > Thank you again for your thoughtful feedback. We clarified in the paper that the main contribution is a decoupled inference framework for improving inference speed, and we’ve addressed your questions about the pruning ratio and scaling behavior. We believe these updates have helped sharpen the contribution.
> >
> > Since the COLM rebuttal period ends soon (June 10), we wanted to follow up in case you had any further thoughts or questions. We’d be happy to continue the discussion.

---

### Official Review · Reviewer_paUo · 2025-05-13

**Rating:** 6
**Confidence:** 3
**Ethics Flag:** 1

**Summary:**

The paper introduces OverFill, a two-stage language model inference framework that leverages a full-size model for the compute-bound prefill stage and a pruned model for the memory-bound decode stage. This architecture achieves improved efficiency and generation quality by balancing computational costs and maintaining performance, showing significant gains over standalone pruned models with minimal latency overhead.

**Questions To Authors:**

1. How does OverFill interact with or complement existing model compression methods like quantization and distillation? Could the prefill stage be quantized more aggressively?
2. You mention that OverFill may underperform for extremely long sequences and suggest refreshing the prefill. Can you elaborate on how this would be implemented practically without breaking autoregressive continuity?
3. Have you tested or do you anticipate OverFill working with mixture-of-expert models e.g. deepseek, llama4, qwen3?

**Reasons To Accept:**

1. The main idea of splitting inference into compute-bound and memory-bound stages and optimizing each separately is well-motivated and practically significant.
2. The empirical results are strong. OverFill consistently outperforms pruned baselines and tuned models across a diverse range of benchmarks (ARC, MMLU, GSM8K, etc.), often achieving performance comparable to larger models.
3. The authors effectively showcase OverFill's balance of latency and accuracy, particularly under long generation and high-batch regimes.

**Reasons To Reject:**

1. While OverFill performs well on standard benchmarks, the performance gap narrows as generation length increases. The authors acknowledge this but don't offer concrete mitigation strategies beyond future work.
2. The approach is validated only on small scale models (<= 8b). It would be more informative to at least evaluate on middle scale models e.g. Llama 3.1 70b or Qwen 2.5 32b or 72b models.
3. The approach relies on partial training of the pruned decoder. However, detailed analysis on training cost savings (vs. full finetuning) is limited.

---

> ### Author Response · Authors · 2025-06-03
>
> Thank you for the constructive feedbacks and and careful reviews.
>
> ### 1. Bigger models
>
> We have added new results with larger models. Results are shown in the general response.
>
> ### 2. Analysis of training cost
>
> Thanks for the suggestion. We provide an analysis of the training costs (FLOPs) for our three setups. Using a formula for Transformer forward FLOPs per token:
>
> $\text{Forward-FLOPs} \approx L \cdot (4H^2 + 2SH + 2HI), $
>
> where $L$ is the number of transformer blocks, $H$ is the hidden dimension, $I$ is the intermediate MLP dimension, and $S$ is the sequence length. This formula accounts for projection, attention, and MLP operations. For total training cost, this should be multiplied by 2 to account for both forward and backward passes.
>
> We compare three models in the 8B → 3B setup, assuming $S = 1024$, and compute their forward FLOPs per token:
>
> |                 | Name              | # of Params | $L$ | $H$ | $I$ | Forward-FLOPs / Token |
> | --------------- | ----------------- | ----------- | ----- | ----- | ----- | --------------------- |
> | Full model      | Llama 3.1 8B      | 8.0B        | 32    | 4096  | 14336 | $F = 6.17B$         |
> | Pruned model    | Llama 3.1 8B → 3B | 3.2B        | 32    | 2432  | 7680  | $f = 2.11B$         |
> | Finetuned model | Llama 3.2 3B      | 3.2B        | 28    | 3072  | 8192  | $f' = 2.64B$        |
>
> We consider three training setups: OverFill, Pruned, and Finetuned. For simplicity, assume each sequence is evenly split between prompt and generation. While this slightly affects the $S$-dependent term, the difference is minor and ignored for clarity.
>
> * **OverFill** processes the first half of tokens with the full model (forward only), and the second half with the pruned model (forward + backward). The full model is frozen, so the total training FLOPs are:
>
> $FLOPs_{overfill} = (S/2) \cdot F + 2 \cdot (S/2) \cdot f \approx 5.20B \cdot S$
>
> * **Pruned** uses the pruned model for the entire sequence:
>
> $FLOPs_{pruned} = (S/2) \cdot f + 2 \cdot (S/2) \cdot f = 3.17B \cdot S$
>
> * **Finetuned** uses the finetuned model similarly:
>
> $FLOPs_{finetuned} = (S/2) \cdot f' + 2 \cdot (S/2) \cdot f' = 3.96B \cdot S$
>
> Thus, OverFill has approximately **1.64×** and **1.31×** higher theoretical training cost compared to the pruned and finetuned models, respectively. In practice, OverFill is roughly 2× slower in runtime during training.
>
> However, we would like to emphasize that our primary focus is **inference efficiency**, as training is a one-time cost.
>
>
> ### 3. Compatibility with existing model compression methods (quantization, distillation)
>
> This is also a good point. We believe OverFill is highly compatible with distillation, whether at the sequence level (e.g., using sampled outputs instead of teacher forcing) or the token level (e.g., learning from the full model’s output logits). Such techniques can benefit both OverFill and the baselines, which we think is orthogonal improvements to the core idea of OverFill. Also, post-training quantization techniques can be applied independently to the prefill and decode models. However, we have not yet evaluated the robustness of our models to quantization.
>
> An interesting future direction would be to prefill with a full-precision model and decode with a quantized model, which may require quantization-aware training. We leave this as future work.
>
>
> ### 4. Elaboration on prefill refresh
> To clarify the idea we mentioned in future work: this direction can be implemented in a way similar to speculative decoding, where a bigger model and a small model are used together during decoding.
>
> Specifically, decoding would proceed as follows:
>
> 1. Prefill the KV cache using the full model with the initial prompt.
> 2. Generate N tokens using the small model (decode model).
> 3. Feed the newly generated N tokens into the full model to refresh its KV cache.
> 3. Repeat steps 2–3 until the sequence is complete.
>
> This approach queries the full model every N tokens instead of only once at the beginning. We believe it can preserve the benefit of "smart prefill" throughout generation, without breaking autoregressive continuity.
>
> The main difference from speculative decoding is that we do not reject the tokens generated by the small model. As a result, this approach avoids verification and rollback steps, which may lead to higher throughput.

---

> > ### Author Response · Authors · 2025-06-03
> >
> > ### 5. MoE models
> > This is a great suggestion. We have not yet tested our method on MoE models, but the most straightforward extension would be to apply width pruning to each expert (MLP) for decoding. We also think OverFill could be combined with expert pruning, as recent work [1, 2] has shown it can improve efficiency by reducing memory usage.
> >
> > [1] Lu, Xudong, et al. "Not all experts are equal: Efficient expert pruning and skipping for mixture-of-experts large language models." arXiv preprint arXiv:2402.14800 (2024).
> >
> > [2] Liu, Enshu, et al. "Efficient expert pruning for sparse mixture-of-experts language models: Enhancing performance and reducing inference costs." arXiv preprint arXiv:2407.00945 (2024).

---

> > > ### Author Response · Authors · 2025-06-09
> > >
> > > Hi Reviewer paUo,
> > >
> > > Thank you again for your helpful review. We added experiments with a larger model (Qwen 14B) and included analysis on OverFill’s training cost, as you suggested. We also addressed your questions about prefill refresh, compatibility with other model compression methods, and applicability to MoE models. We believe these additions have improved the paper.
> > >
> > > Since the COLM rebuttal period ends soon (June 10), we wanted to follow up in case you had any further thoughts or questions. We’d be happy to continue the discussion.

---

### Author Response · Authors · 2025-06-03

We appreciate the reviewers' constructive feedback and insightful questions. In response, we have added new results and clarifications to strengthen the paper.

We provide new results using larger models (14B for prefill and 3B or 7B for decode) from a different model family (Qwen). Due to time constraints, these models were trained on a smaller dataset (OpenHermes 2.5), which is approximately 1/7 the size of the dataset used in our main experiments.

Despite the lighter training, OverFill consistently outperforms the pruned model across all benchmarks. It also achieves stronger performance than similarly sized finetuned models on shorter-length benchmarks. We have seen that performance on longer-length benchmarks improving steadily as training progresses, and will include the full training results in the final version of the paper.

### **Qwen 2.5 14B -> 3B**

| Model     | ARC   | GSM8K | Ifeval (inst level) | Ifeval (prompt level) | MMLU  | WMT16-DE-EN |
|-----------|-------|--------|----------------------|------------------------|--------|---------------|
| Overfill  | **90.6** | 40.3   | **54.2**              | **43.4**                | **77.9** | 31.3          |
| Pruned    | 28.7  | 13.3   | 26.3                 | 15.7                   | 26.4   | 7.1           |
| Finetuned | 79.5  | **72.9** | 46.1                 | 33.2                   | 62.2   | **37.1**       |

### **Qwen 2.5 14B -> 7B**


| Model     | ARC   | GSM8K | Ifeval (inst level) | Ifeval (prompt level) | MMLU  | WMT16-DE-EN |
|-----------|-------|--------|----------------------|------------------------|--------|---------------|
| Overfill  | **90.6** | 71.5   | **64.0**              | **53.6**                | **77.9** | 37.2          |
| Pruned    | 66.7  | 64.3   | 45.3                 | 33.4                   | 49.4   | 31.0          |
| Finetuned | 86.8  | **80.2** | 47.4                 | 33.6                   | 69.3   | **38.9**       |

---

### Author Response · Authors · 2025-06-08

Hello ACs and reviewers,

Thank you again for your reviews, which helped us improve the paper.

We updated our paper with:
* New experiments using larger models (14B for prefill) and additional model families (Qwen),
* Analysis of training cost,
* Benchmarking results comparing our method to speculative decoding, and
* Clarifications to the motivation and main figure, as suggested.

We’re aware that the COLM rebuttal period ends in three days (June 10). We would welcome the opportunity for another round of feedback. Please let us know if you have any further questions.

Warm regards,

Authors of Overfill

---

### Decision · Program_Chairs · 2025-07-08

**Decision:**

Accept

**Comment:**

The reviews are lukewarm about this paper, sitting just slightly above borderline. I believe there are genuine concerns about experiments such as scale of experiments, and fully validating claims through from-scratch trained of a pruned model. Going beyond LLaMa was done during rebuttal time. In general the authors have put a lot of work on demonstrating this efficiency technique, unfortunately efficiency techniques are expensive to validate for large language models without full scale experiments. Reviewers have largely overlooked the rebuttal so I think while some of the concerns are legitimate as AC I have looked at both points and I am also slightly over the fence for acceptance as some of the ideas on this work might survive scrutiny when tested at scale, particularly the idea of a prefill and an overfill model for decoder-only architectures.